# Measurement of Slips at Contact Interfaces Using a Self-Powered Sensor Based on Triboelectric Nanogenerators

**DOI:** 10.3390/nano12193510

**Published:** 2022-10-07

**Authors:** Wangjia Zhao, Wenjie Qin, Mingsen Ba, Yu Sun

**Affiliations:** School of Mechanical Engineering, Beijing Institute of Technology, Beijing 100081, China

**Keywords:** TENG, self-powered sensor, slip, contact

## Abstract

The accumulation of tangential small slips at contact interfaces may occur in mechanical assemblies when subjected to cyclic loadings, and cause failure of parts. However, the slip is difficult to measure directly. In this paper, a self-powered displacement sensor based on triboelectric nanogenerator (TENG) is fabricated, which is used to measure the microslip of the specimen in the designed test bench. When the specimen slips under the cyclic tangential load with a frequency of 10 Hz, the resulted average slip displacement of the specimen under a single load cycle is 9.01 μm, which is measured by the Keithley DAQ 6510 multimeter (Tektronix, Shanghai, China). This value is close to that measured by the Keithley 6514 electrometer. In addition, adding a ${\mathrm{BaTiO}}_{3}$ thin film with a thickness of 100 nm between the Kapton triboelectric layer and the Cu electrode, the output voltage of the sensor is increased by 51.5%. Thus, the sensitivity of the self-powered sensor is improved effectively.

## 1. Introduction

There are many fastening structures in mechanical assemblies connected by frictional contacts. When they are subjected to cyclic tangential loads, frictional slips may occur along the interfaces, and the accumulation of slips may further lead to a significant global relative displacement between parts, such as the rotation of a bush in the small end of a connecting rod and loosening of bolted joints [1,2]. Given that slips will cause loosening of fastening connections, and even system failures, measuring the slip displacement between contact surfaces is of great significance to the safety of mechanical assemblies.

Sensors commonly used to measure displacement include eddy current sensors, laser sensors, LVDT sensors and so on [3,4,5]. Among them, a LVDT sensor is often used in fretting experiments to measure the tangential displacement between contact specimens as their slip. However, this displacement is not the same as the slip in the contact, because there are components with compliance which physically separate the contact from the point of measurement, and hence, the measured displacement amplitude is always larger than the contact slip amplitude [5]. In fact, the slip at different positions at the contact interface is different, and it is difficult to measure the slip at a certain position directly with the above sensors. In addition, these sensors need external energy during testing.

A triboelectric nanogenerator (TENG) can convert mechanical energy into electrical energy, and its output electrical signal can be used to analyze the mechanical motion in turn [6], so it can be used as a self-powered sensor. At present, a variety of self-powered displacement sensors based on the principle of TENG have been developed [7,8,9,10]. For example, Zhou et al. designed a one-dimensional displacement sensor composed of a pair of micro grating structures [11]. With the sliding of the measured object, the grating will be separated and contacted continuously, and the displacement of the object is obtained by calculating the number of voltage peaks and valleys. Given that grating-structured TENGs based on interdigital electrode structures are unable to sense the motion direction of the freestanding electrode in real time, Li et al. designed a self-powered vector/displacement sensor (SPVS) composed of one group of grating freestanding electrodes and three groups of interdigitated introduction electrodes with the identical period, which can distinguish the real-time direction of the freestanding electrode displacement [12]. Yang et al. fabricated a self-powered rotation speed and displacement sensor using the dual cylinder structure TENG to measure the displacement of the rotating mobile devices [13]. It is composed of two cylinders and rotates with the rolling of the measured object, generating voltage which is used to record the number of turns. The displacement of the measured object can be obtained by multiplying the number of turns by the length of the rotation object. However, these self-powered sensors based on the principle of TENG are often used to measure the macroscopic displacement of objects, and there are few reports on measuring the microslip between contact surfaces. We do not know whether this kind of self-powered sensor is sensitive enough to measure the micro contact slip and its accumulation. If it can, it can be applied to monitor the loosening failure of fastening connection structures.

In this paper, a self-powered sensor based on TENG is fabricated to measure the contact slips directly in the test bench built in this study. Fixing the two triboelectric layers of the sensor on the upper and lower specimen respectively, the slip displacement of several microns can be measured. Unlike an electrometer, such as a Keithley 6514 electrometer, which is widely used to obtain the electrical signal output of a TENG in many studies, a multimeter (Keithley DAQ 6510) is adopted to measure the output voltage of the sensor, which can obtain the close slip values compared with the results measured by the Keithley 6514 electrometer. Furthermore, in order to improve the sensitivity of the self-powered sensor, a BaTiO3 thin film is added between the triboelectric layer and the electrode to enhance the output voltage of the self-powered sensor.

## 2. Materials and Methods

### 2.1. Principle of Self-Powered Sensor

There are four working modes of TENG: contact-separation mode [14], sliding mode [15], single-electrode mode [16], and freestanding triboelectric-layer-based mode [17]. In this study, the contact-separation mode and sliding mode are considered.

The principle of a contact-separation TENG is the coupling of contact electrification and electrostatic induction [18]. After the two triboelectric layers with different materials being forced to contact with each other, charges with the same density and opposite sign will be induced on the contact surfaces. When the two triboelectric layers are separated under the force, electric potential difference is then established between the two electrodes which are connected to the triboelectric layers. If the area size of the metal electrodes is much larger than their separation distance, the relationship between the open circuit voltage ($V_{\mathrm{OC}}$) generated by the TENG and the separation distance (x(t)) is [19]:(1)$$V_{OC}=\frac{\sigma x\left(t\right)}{\varepsilon_{0}},$$
where σ is the charge density and $\varepsilon_{0}$ is the permittivity of vacuum.

As for a sliding-mode TENG, when the two contacting triboelectric layers slide relatively with a distance *x*(*t*) << *l* in which *l* is the length of the triboelectric layer, electric potential difference is also established between the two electrodes due to the reduction of contact area. The relationship between the open circuit voltage ($V_{\mathrm{OC}}$) and the sliding distance (x(t)) is [20]:(2)$$V_{OC}=\frac{\sigma x\left(t\right)d_{0}}{\varepsilon_{0}\left[l-x\left(t\right)\right]},$$
where and $d_{0}$ is effective thickness constant, which is defined as:(3)$$d_{0}=\overset{n}{\underset{i=1}{\sum }}\frac{d_{i}}{\varepsilon_{ri}},$$
where $d_{i}$ is the thickness of the triboelectric layer and $\varepsilon_{\mathrm{ri}}$ is the permittivity of the triboelectric layer.

By comparing Equations (1) and (2), it can be seen that when *x* << *l*, the open circuit voltage of the contact-separation TENG is much greater than that of the sliding-mode TENG, and the open circuit voltage of the contact-separation TENG is directly proportional to the separation distance between the two electrodes. In order to ensure that the sensor can obtain larger electrical signals and better linearity, the contact-separation TENG is used as the self-powered sensor in this study.

### 2.2. Test Bench

The test bench designed in this experiment is shown in Figure 1. The upper specimen is in contact with the lower specimen under the uniformly distributed pressure which is applied by the steel cube. The positive triboelectric layer and the negative triboelectric layer of the self-powered sensor are fixed on the upper specimen and the lower specimen respectively, and perpendicular to the slip direction. The tangential cyclic load generated by the vibration exciter is applied to the upper specimen through a rod. A force sensor is mounted on one end of the rod to measure the exciting force on the rod. The preload is measured by another force sensor in the vibration exciter. When the cyclic load is great enough, the upper specimen begins to slide, resulting a separating distance between the positive and negative triboelectric layers of the sensor. The output electrical signal of the sensor is collected by a Keithley DAQ 6510 multimeter (Tektronix, Shanghai, China) and the slip displacement can be calculated from the measured voltage.

The self-powered sensor is a conductor-to-dielectric TENG, working in the contact-separation mode as shown in Figure 2. The size of each triboelectric layer is 25 mm×10 mm. One triboelectric layer is a 0.06 mm copper film (also used as electrode), and the other triboelectric layer is a 0.06 mm Kapton film with a copper film as the other electrode. Both triboelectric layers are bonded to the acrylic substrates with a thickness of 2 mm. Two conducting wires connected with the two electrodes are accessed to the Keithley DAQ 6510 multimeter.

### 2.3. Signal Analysis and Test Instrument

At present, a Keithley 6514 electrometer is widely used to measure the voltage signal generated by the TENG [21,22]. In this study, the voltage signal of the self-powered sensor is measured by a multimeter, the Keithley DAQ 6510 multimeter, which can also achieve satisfactory accuracy. The equivalent circuit of the system including the TENG and the resistance load is shown in Figure 3 and its equation can be obtained by using Kirchhoff’s law, as shown in Equation (4) [23].
(4)$$R\frac{dQ}{dt}=-\frac{Q}{C}+V_{OC},$$

In Equation (4), *Q* is the transferred charge. Setting *Q*(*t* = 0) = 0, Equation (4) can be solved as and can be calculated as:(5)$$Q\left(t\right)=\frac{1}{R}exp\left[-\frac{1}{R}\int_{0}^{t}\frac{1}{C\left(x\left(t\right)\right)}dt\right]\int_{0}^{t}V_{OC}\left(x\left(t\right)\right)exp\left[\frac{1}{R}\int_{0}^{t}\frac{1}{C\left(x\left(t\right)\right)}dt\right]dt$$

As $I\left(t\right)=\frac{dQ\left(t\right)}{dt}$, the current in the system is:(6)$$I\left(t\right)=\frac{V_{OC}}{R}-\frac{1}{R^{2}C}exp\left[-\frac{1}{R}\int_{0}^{t}\frac{1}{C\left(x\left(t\right)\right)}dt\right]\int_{0}^{t}V_{OC}\left(x\left(t\right)\right)exp\left[\frac{1}{R}\int_{0}^{t}\frac{1}{C\left(x\left(t\right)\right)}dt\right]dt$$

Then, the voltage of the resistance *R* is:(7)$$V\left(t\right)=V_{OC}-\frac{1}{RC}exp\left[-\frac{1}{R}\int_{0}^{t}\frac{1}{C\left(x\left(t\right)\right)}dt\right]\int_{0}^{t}V_{OC}\left(x\left(t\right)\right)exp\left[\frac{1}{R}\int_{0}^{t}\frac{1}{C\left(x\left(t\right)\right)}dt\right]dt$$

If the separation distance of the electrodes is kept constant in the TENG, that is, x(t) is a constant *S*, the output voltage is:(8)$$V\left(t\right)=V_{OC}e^{-\frac{t}{RC}},$$
which decays in the form of negative exponential function.

In the Equations (8), the maximal output voltage $V_{\mathrm{OC}}$ is the open circuit voltage when the separation distance of the electrodes is *S*. Usually, the peak voltage measured by a multimeter (as shown in Figure 4) cannot reach the open circuit voltage of TENG. According to the study on the influence of the load resistance on the magnitude of the output voltage by Niu [23], it can be seen that if the resistance is greater than 1 GΩ, the output voltage is close to the open circuit voltage. Therefore, the resistance value of the multimeter is set to 10 GΩ in this study, and the measured peak voltage is close to the open circuit voltage $V_{\mathrm{OC}}$.

## 3. Results

### 3.1. Calibration of the Sensor

The two triboelectric layers are separated by a distance of 0.8 mm then kept for a period of time for several times, and the measured voltage results are shown in Figure 4. According to the analysis in Section 2.3, the peak voltage value is close to the open circuit voltage $V_{\mathrm{OC}}$ of the sensor, which is 3.59 V on average.

It can be seen that the measured voltage decreases exponentially with time after it gets to the peak. This will cause a serious distortion in voltage measuring at a low-frequency condition. As the excitation frequency of the cycle load in this study, 10 Hz, is concerned, the upper specimen slides every 0.1 s, and the measured voltage will decay for 0.1 s before the next slip. It can be deduced from Figure 4 that the attenuation is about 1/100 (from 3.59 V to 3.56 V), which can be ignored in the single slip measurement.

According to Equation (1), the open circuit voltage has a linear relationship with the separation distance of the self-powered sensor, so the relationship between the measured voltage *V* (V) and separation distance *x* (mm) is:(9)V=4.49x

### 3.2. Dynamic Slip Test Results of Contact Surface under Cyclic Load

Setting the frequency of cycle load as 10 Hz, the time-varied cyclic load is applied to the upper specimen, and the change of the amplitude of the cyclic load is shown in Figure 5. When the time is about 81 s, the sensor starts to output voltage, that is, the upper specimen begins to slip. At this time, the amplitude of the cyclic load is 11.84 N, and the preload is 1.5 N. Therefore, the maximal force acting on the upper specimen is 13.34 N, which is the load to overcome the friction. The whole slip process lasts 7 s (from 81 s to 88 s). After that, the load is withdrawn.

According to the voltage value generated by self-powered sensor, and considering the relationship between the measured voltage and separation distance, the resulted slip curve of the upper specimen under the action of excitation force is shown in Figure 6. As the load is withdrawn, the measured voltage begins to drop. From the detail of the slip from the time of 81.15 s to 82.15 s, a total sliding distance of 90.13 μm for the 10 load cycles at the initial stage of slip is obtained, and the average slip value of each load cycle is 9.01 μm.

### 3.3. Verification of the Test Accuracy Using Keithley 6514 Electrometer

In order to investigate the accuracy of the above test results, the slip displacement as the same conditions is measured by the Keithley 6514 electrometer.

The two triboelectric layers are also separated by 0.8 mm then kept for a period of time for several times, and the measured voltage results are shown in Figure 7. Different from the voltage change measured by the Keithley 6510 multimeter, the voltage change measured by the Keithley 6514 electrometer is basically unchanged after reaching its peak value during the period when the triboelectric layers are separated by 0.8 mm and remains unchanged. The reason is that there is almost no charge transfer in the circuit during this period. The greatest value of the voltage is close to the open circuit voltage $V_{\mathrm{OC}}$ of the sensor. It can be seen from the figure that the maximum voltage value is 3.50 V on average. Taking this value as the open circuit voltage of the self-powered sensor, the relationship between the measured voltage *V* (V) and the separation distance *x* (mm) is:(10)V=4.375x

The frequency and amplitude of the cycle load is the same as in Section 3.2, and the slip curve of the upper specimen obtained by the dynamic slip test is shown in Figure 8. From the figure, it can be seen that the slip process lasts for 7 s (from 77.5 s to 84.5 s). After that, the load is withdrawn and the measured voltage remains constant. From the detail figure of the slip, a total sliding displacement of 94.50 μm for the 10 load cycles at the initial stage of slip is obtained, and the average slip value of each load cycle is 9.45 μm, which is close to the result measured by the Keithley 6510 multimeter.

Comparing the measurement results of the Keithley DAQ 6510 multimeter and that of the Keithley 6514 electrometer, it is found that the test result of Keithley DAQ 6510 multimeter is close to that of Keithley 6514 electrometer, so it is feasible to obtain the voltage of the self-powered sensor used to measure the dynamic slip at this frequency with a multimeter such as the Keithley DAQ 6510 multimeter.

## 4. Performance Enhancement of Sensor by ${\mathrm{BaTiO}}_{3}$ Thin Film

### 4.1. Role of ${\mathrm{BaTiO}}_{3}$ Thin Film 

According to Equation (1), the open circuit voltage of the contact-separation sensor is proportional to the charge density if the separation distance is constant. Therefore, the measured voltage of the self-powered sensor can be improved by increasing the charge density. Since ${\mathrm{BaTiO}}_{3}$ is a ferroelectric ceramic material with dielectric polarization, it is known to have and residual dielectric polarization after being exposed to an electrical field. When the dielectric polarization and the surface polarization of the triboelectric layer are coupled, the residual polarization of ${\mathrm{BaTiO}}_{3}$ can enhance the charge density, thereby increasing the open circuit voltage of the sensor [24]. Thus, a ${\mathrm{BaTiO}}_{3}$ thin film is added between the triboelectric layer and the electrode in this study.

The ${\mathrm{BaTiO}}_{3}$ thin film is deposited by an RF (radio frequency) magnetron sputtering on the copper electrode from the ceramic target. The sputtering time is 3 h and the obtained film thickness is 100 nm. It is placed between the Kapton layer and the copper electrode of the sensor in Section 2.2, as shown in Figure 9, and its SEM image is shown in Figure 10.

### 4.2. Measurement Results Using the Self-Powered Sensor with ${\mathrm{BaTiO}}_{3}$ Thin Film

The self-powered sensor with a ${\mathrm{BaTiO}}_{3}$ thin film is calibrated firstly by the same method in Section 3.1, and the measured voltage when the two triboelectric layers are separated by a distance of 0.8 mm is shown in Figure 11. The peak voltage is 5.44 V, so the relationship between the measured voltage *V* (V) and separation distance *x* (mm) is:(11)V=6.8x

Setting the frequency and amplitude of the time-varied cyclic load is consistent with Section 3.2, and the slip curve of the upper specimen obtained by the dynamic slip test is shown in Figure 12. It can be seen that the slip process lasts for 7 s and the total displacement within 1 s at the initial stage of sliding is 89.41 μm from the slip details. Thus, the average slip displacement value of each cyclic load is 8.94 μm, which is close to that obtained in Section 3.

Comparing the measurement results of the self-powered sensor with ${\mathrm{BaTiO}}_{3}$ thin film and that of the self-powered sensor without the ferroelectric materials, the measured voltage value of the self-powered sensor with ${\mathrm{BaTiO}}_{3}$ thin film is increased by 51.5%; thus, the sensitivity of the self-powered sensor is effectively improved.

## 5. Conclusions

A self-powered sensor is fabricated to measure the contact slips directly in the test bench built in this study, which is a conductor-to-dielectric TENG with one triboelectric layer is a 0.06 mm copper film and the other triboelectric layer is a 0.06 mm Kapton film. The sensor can directly measure the slip displacement of several microns of the contact surface in the test bench. When the specimen slips under the cyclic tangential load with a frequency of 10 Hz, the resulted average slip displacement of the specimen under a single load cycle measured by the Keithley DAQ 6510 multimeter is 9.01 μm, which is close to the slip value measured by the Keithley 6514 electrometer, that is 9.45 μm. Thus, a multimeter can be used to obtain the output voltage of the self-powered sensor, which is used to measure the dynamic slip at this frequency.

In addition, a ${\mathrm{BaTiO}}_{3}$ thin film with a thickness of 100 nm is added between the Kapton layer and the Cu electrode of the sensor. The measured voltage is increased by 51.5% compared with the self-powered sensor without ${\mathrm{BaTiO}}_{3}$ thin film. Thus, the sensitivity of the sensor can be effectively improved by adding a ${\mathrm{BaTiO}}_{3}$ thin film under the triboelectric layer.

## Figures and Tables

**Figure 1 nanomaterials-12-03510-f001:**
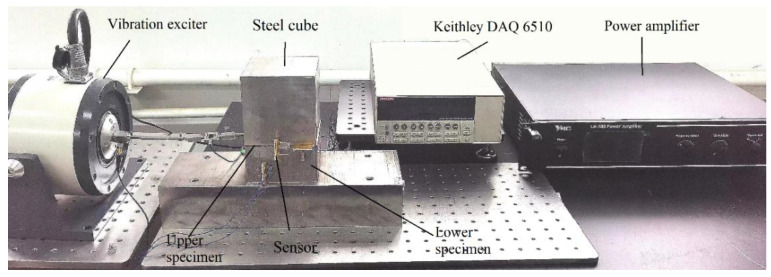
The test bench.

**Figure 2 nanomaterials-12-03510-f002:**
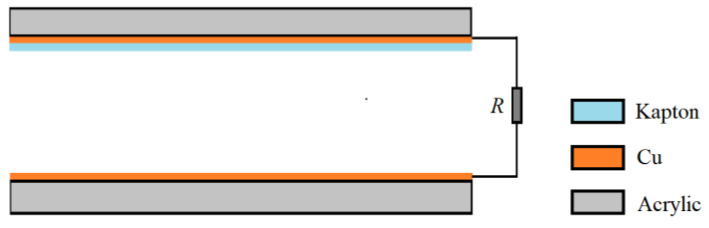
Structure of the self-powered sensor.

**Figure 3 nanomaterials-12-03510-f003:**
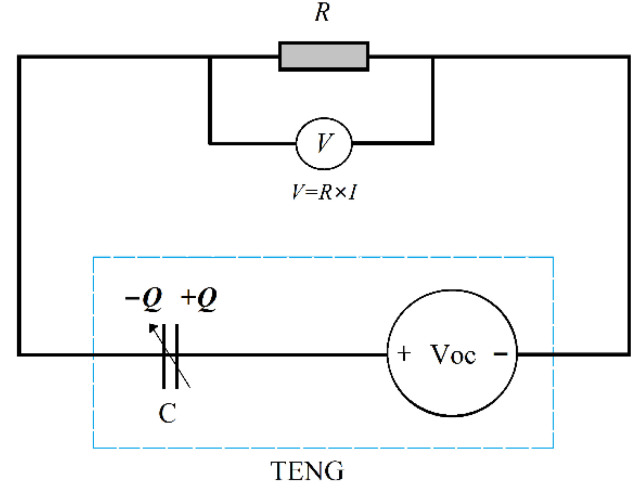
Equivalent circuit.

**Figure 4 nanomaterials-12-03510-f004:**
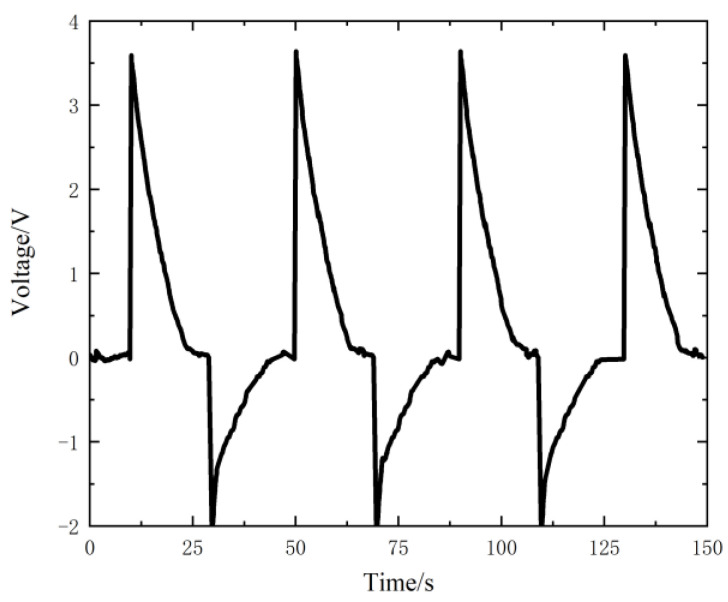
Voltage output of the sensor measured by the Keithley 6510 multimeter.

**Figure 5 nanomaterials-12-03510-f005:**
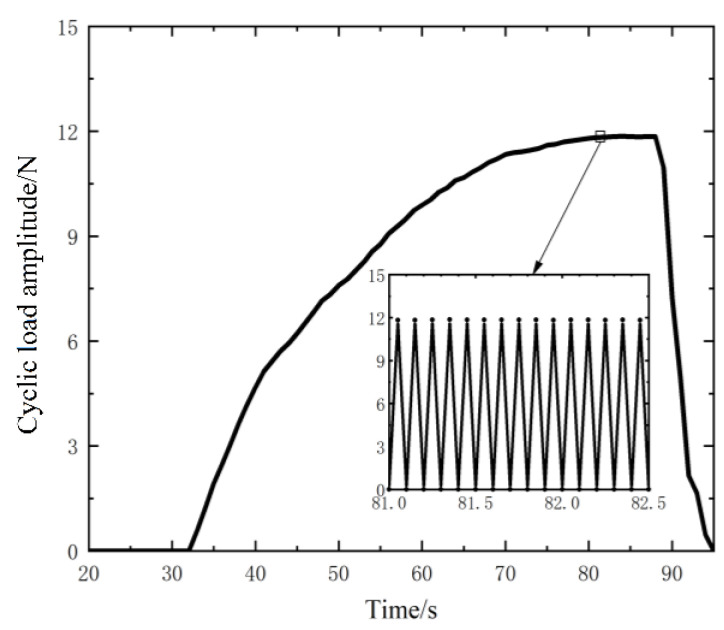
The amplitude of the cyclic load.

**Figure 6 nanomaterials-12-03510-f006:**
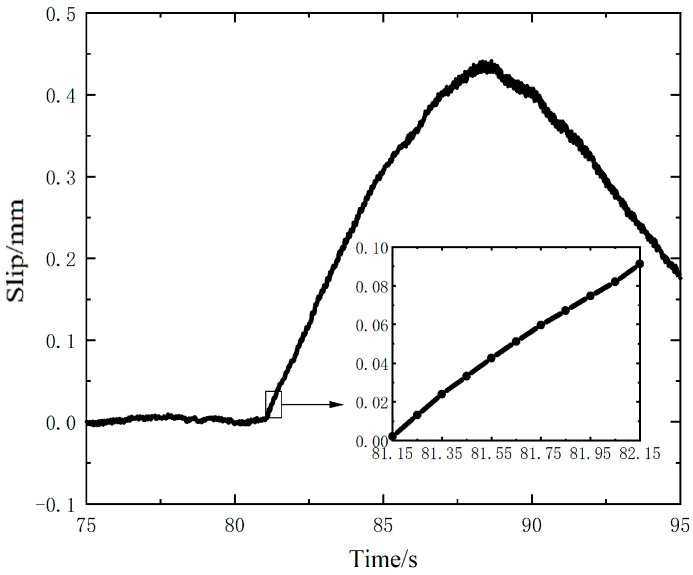
Slip measured by the sensor using the Keithley 6510 multimeter.

**Figure 7 nanomaterials-12-03510-f007:**
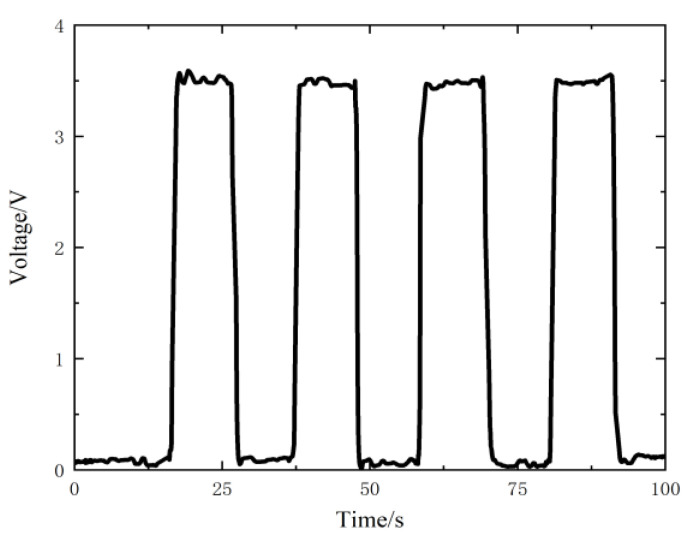
Voltage output of the sensor measured by the Keithley 6514 electrometer.

**Figure 8 nanomaterials-12-03510-f008:**
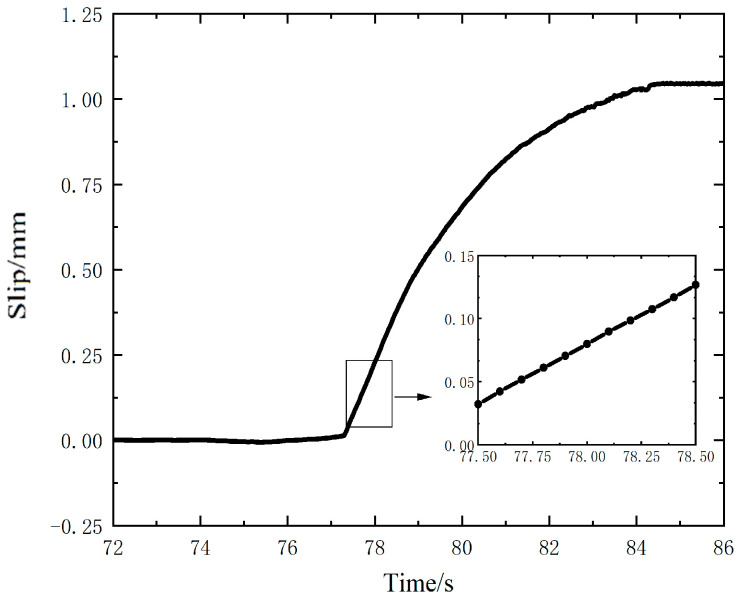
Slip measured by the sensor using the Keithley 6514 electrometer.

**Figure 9 nanomaterials-12-03510-f009:**
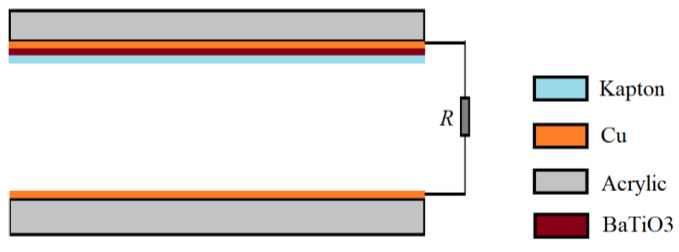
Structure of the sensor with a ${\mathrm{BaTiO}}_{3}$ thin film.

**Figure 10 nanomaterials-12-03510-f010:**
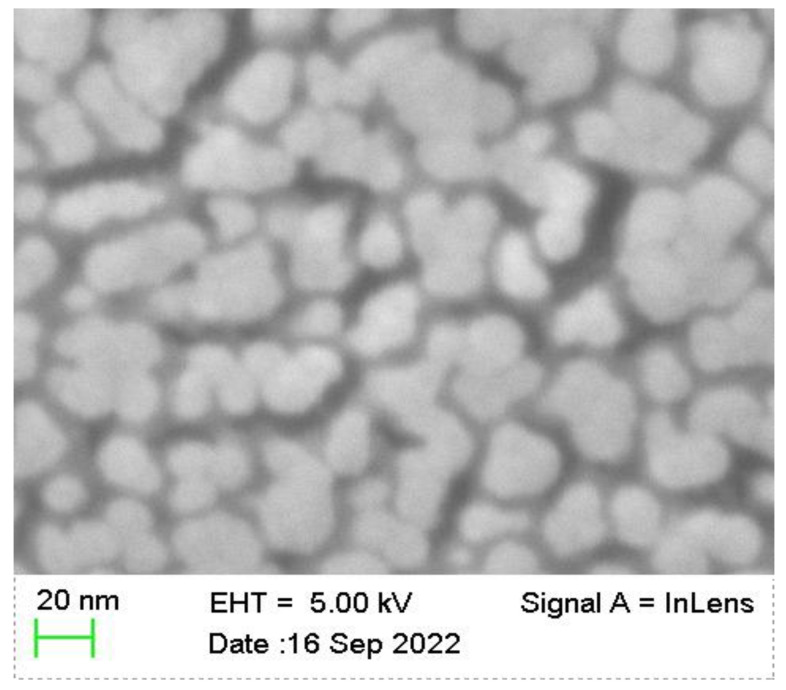
SEM of the ${\mathrm{BaTiO}}_{3}$ thin film.

**Figure 11 nanomaterials-12-03510-f011:**
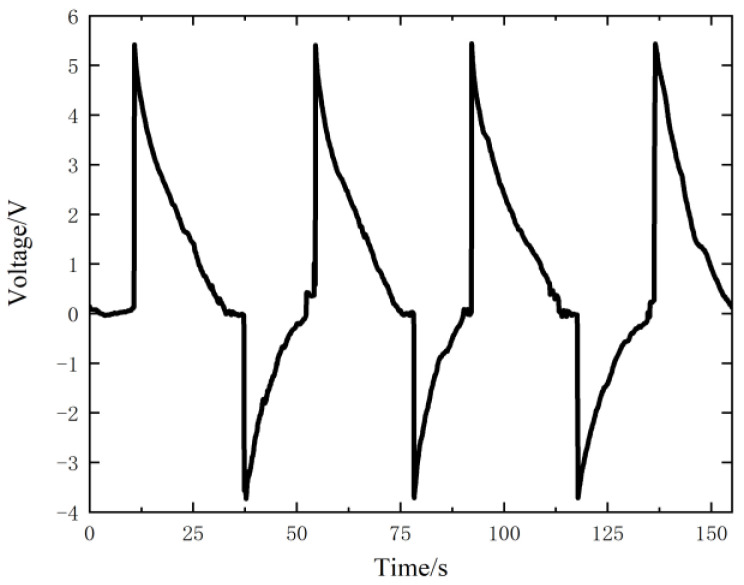
Voltage output of the sensor with a ${\mathrm{BaTiO}}_{3}$ thin film.

**Figure 12 nanomaterials-12-03510-f012:**
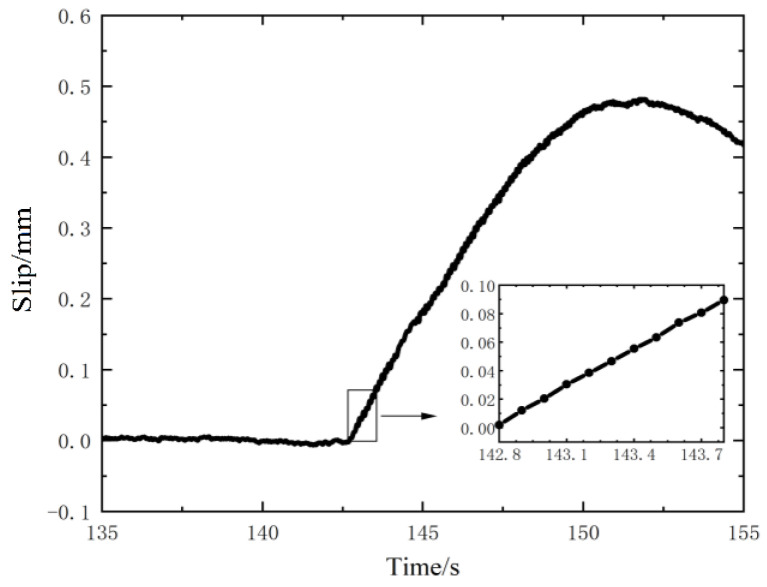
Slip measured by the sensor with a ${\mathrm{BaTiO}}_{3}$ thin film.

## Data Availability

The data are available on reasonable request from the corresponding author.

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
