# Peer review of "Measurement of Slips at Contact Interfaces Using a Self-Powered Sensor Based on Triboelectric Nanogenerators"

_nanomaterials, 2022, doi:10.3390/nano12193510_

Round 1
Reviewer 1 Report
In this paper, the authors reported a self-powered displacement sensor based on triboelectric nanogenerator (TENG) to measure the slips at contact interfaces. It is recognized that measurement of slips has various practical applications, however, I don't think this work is sufficiently innovative for publication in its current version.
Detailed comments:
1) The author stated that "However, these self-powered sensors based on the principle of TENG are often used to measure the macroscopic displacement of objects, and there are few reports on measuring the microslip between contact surfaces." It seems that 1) the measurement of micro slip is just another application of micro-displacement measurement. and 2) it seems that some research exists on measuring the microslip. The authors need to justify the innovation and state-of-the-art.
2) Another result that emphasized is "multimeter vs. electrometer". Again, there is no justification of one over the other.
3) There is no comparison of this measuring results to other methods. It is unclear what is the advantages of this methods.
Reviewer 2 Report
Present Manuscript discuss about the sensor application of TENG device. Article is interesting and well written. However method section should be improved.
(1) Authors used BaTiO3 film as an active layer in the TENG device. However no preparation method of the film is given. Authors must clarify nature of the film i.e. if it is ceramic or polymer composite film.
(2) SEM micrograph shows scale bar 500 nm and yet the grain contrast is very well. We expect for a 100 nm thick film, grains must be very small to be resolved at that scale bar.
Round 2
Reviewer 1 Report
The authors addressed my comments in the revised manuscript. I would suggest accept this work.